# Severe COVID-19 in Cardiopath Young Pregnant Patient without Vertical Transmission

**DOI:** 10.3390/v14040675

**Published:** 2022-03-25

**Authors:** Ana Paula Figueiredo de Montalvão França, Danielly do Vale Pereira, Elaine Valéria Rodrigues, Flávia Nunes Vieira, Karine Santos Machado, Pedro Aleixo Nogueira, Ricardo Roberto de Souza Fonseca, Luiz Fernando Almeida Machado

**Affiliations:** 1Biology of Infectious and Parasitic Agents Post-Graduate Program, Federal University of Pará, Belém 66075-110, PA, Brazil; anapaula@alexandrohup.com (A.P.F.d.M.F.); ricardofonseca285@gmail.com (R.R.d.S.F.); 2Virology Laboratory, Institute of Biological Sciences, Federal University of Pará, Belém 66075-110, PA, Brazil; 3Santa Casa de Misericórdia do Pará Foundation, Belém 66075-110, PA, Brazil; daniellydovale@hotmail.com (D.d.V.P.); elainerik@gmail.com (E.V.R.); flaviavieira99@gmail.com (F.N.V.); karinemachado18@outlook.com (K.S.M.); 4School of Medicine, University Center of State of Pará, Belém 66060-575, PA, Brazil; nogueirapedro94@gmail.com

**Keywords:** SARS-CoV-2, cardiological infection, congenital transmission, coronaviruses

## Abstract

The new betacoronavirus, known as severe acute respiratory syndrome coronavirus 2 (SARS-CoV-2), is the cause of COVID-19, and has spread rapidly around the world, reaching more than 200 countries, around 364 million people and causing more than 5 million deaths according to the World Health Organization, so this paper reports a fatal case of SARS-CoV-2 infection in a young pregnant woman with heart disease, without vertical transmission. A 26 years old patient at 28th week of pregnancy with regular prenatal care, presented dry cough, high fever, and severe respiratory distress. Due to her clinical symptoms, she sought medical care at a cardiology hospital in Northern Brazil. The medical conditions she presented were heart disease, rheumatic fever history and had no recent record of national or international travel. She was hospitalized and after clinical stabilization, she was referred for an emergency cesarean intervention. The young mother and the newborn were transferred to the intensive care unit after surgery, where diagnostic tests for respiratory viral infections, including SARS-CoV-2, were performed. The mother tested positive, while her newborn was negative for SARS-CoV-2 demonstrating no vertical transmission of SARS-CoV-2 in this severe case. This study reveals that despite the mother’s initially mild symptoms, she progressed to severe clinical conditions resulting in death, although no vertical transmission was observed. This report highlights the relevance of comorbidities for the unfavorable clinical course of COVID-19.

## 1. Introduction 

In early December 2019, in Wuhan city, Hubei province, some cases of severe acute respiratory syndrome emerged. This respiratory infection spread rapidly worldwide, leading to great levels of public health concern. The World Health Organization (WHO) named this illness Coronavirus Disease 2019 (COVID-19) and its virus etiological agent SARS-CoV-2 [1,2]. The symptomatic patient presents comprehensive symptoms such as cough, fever, fatigue, loss of smell and taste, and shortness of breath [3].

Epidemiological data identified an increased risk for severe illness of COVID-19, in population groups with cardiovascular diseases, kidney injuries, lung diseases, cancer, diabetes, immunosuppression, hypertension, obesity, or other comorbidities [4,5]. These health risk conditions seem to be involved in the pathogenesis of severe COVID-19 disease, developing severe respiratory symptoms as respiratory distress, severe pneumonia, respiratory failure, and cardiovascular complications that may lead to death [6,7,8].

According to WHO [2], Brazil reported 24,535,884 confirmed cases and 624,413 deaths by SARS-CoV-2 by February 2022. Currently, Brazil is ranked in third place in cumulative deaths by COVID-19 worldwide. Among the obstetric population (pregnant and postpartum patients) who developed severe acute respiratory distress related to infection by SARS-CoV-2, the fatality rate was about 7% in Brazil from February 2020 to January 2022. The main risk factors in this group were obesity, diabetes, and cardiovascular disease [9]. It is due to this fact that the COVID-19 pandemic seriously affects the Brazilian health-care system, especially specialized maternal services, as the availability of structures, medicines and health professionals were directed to the treatment of COVID-19, leaving maternal health services on the sidelines. This report aimed to describe the clinical characteristics of a young pregnant cardiopathic woman who developed a severe case of SARS-CoV-2 infection with death, without vertical transmission of SARS-CoV-2 to the neonate.

## 2. Materials and Methods

### 2.1. Study Design

A 26-year-old female at 28 weeks of pregnancy, with a Body Mass Index (BMI) of 26.6 (height = 160, weight = 68), attending regular prenatal care and with a medical history of heart disease due to rheumatic fever is the subject of this study. She looked for medical care in a cardiology hospital in Belém city, Pará, Northern Brazil, with a ventricular tachyarrhythmia without acute or chronic respiratory infection symptoms. At admission (March 2020), she reported no history of recent national or international traveling, and her medical reports indicated a mitral valve replacement in 2011 due to mitral valve stenosis.

Following initial medical care and clinic stabilization, the patient was discharged 16 days after admission with stable clinical conditions, good fetal health, and no respiratory-infection symptoms. After five days of discharge, she returned to the same hospital seeking medical care with the clinical report of dry cough, high fever (above 39 °C), respiratory distress, tachycardia (>145 Beats Per Minute, BPM), high blood pressure (147/95 mmHg), tachypnea (44 Incursions Per Minute, IPM), and a blood oxygen saturation level of 72%. The patient was undergoing endotracheal intubation and Invasive Mechanical Ventilation (IMV). Samples of blood and nasopharyngeal fluid were collected to screen for the laboratory diagnosis of viral respiratory infections such as influenza A and B, parainfluenza, respiratory syncytial virus, rhinovirus, adenovirus, SARS-CoV-2, and other routine tests.

The patient was taken to the surgery room for an emergency cesarean procedure. The mother and the newborn female baby required intensive care and IMV. A post-operative thoracic Computed Tomography (CT) of the mother’s lung showed multiple ground-glass parenchymal opacities, predominantly in the upper lobes, involving more than 60% of both lungs (Figure 1). On the day after admission and surgery, she began therapy with tazocin + oseltamivir, chloroquine + azithromycin, and dobutamine. The patient evolved to septic shock, which reversed within 24 h.

### 2.2. Clinical Protocol

She presented a rapid clinical improvement on the second day (D2) after surgery. There was a reduction in her major clinical parameters. The temperature was at 37.5 °C, heartbeats 82 BPM, blood pressure 130/80 mmHg, respiratory rate 20 IPM, and oxygen saturation 96%, while the IMV procedure was sustained. The IMV extubation was performed when respiratory stabilization was achieved. On D3, a new blood test was carried out (and performed on a daily routine). The results stated are in Table 1. On D4, the real-time reverse transcriptase-polymerase chain reaction (RT-PCR) assay tested positive for SARS-CoV-2. After the confirmed diagnosis, the patient remained conscious, hemodynamically stable (90 mmHg Invasive Blood Pressure (IBP) and heartbeats 80 bpm), breathing without IMV (oxygen saturation 95%), complaining of persistent dry cough, and mild breath pain. No vasoactive medication was used. The vital signs were at normal ranges despite the oxygen saturation beginning to vary between 93% and 98%. This supplemental oxygen was delivered by a nasal cannula.

On D4 and D5, the patient oscillated between periods of sleepiness and motor agitation. She presented fever episodes, dry cough and periods of ventricular tachyarrhythmia. The ventricular tachyarrhythmia was controlled with amiodarone, administered via a continuous infusion pump. The patient complained of chewing and deglutition difficulties, so she was provided with enteral nutrition. On D6, the medical examination reported moderate somnolence, fatigue, responsiveness to commands, eupneic, 95 mmHg IBP, heartbeats 70 BPM, mitral murmur, O_2_ administered by nasal catheter, blood oxygen saturation 95%, oliguria, dry cough, and leukocytosis up to 12,000/mm^3^. On the night of D7, diarrhea and abdominal discomfort started that led to naso-enteral tube closure.

On D7, the patient’s clinical symptoms worsened. The clinical parameters were tachycardia (132 BPM), hypertension (141/110 mmHg), respiratory difficulty returned, requiring endotracheal intubation and IMV. On D9, norepinephrine and dobutamine were administered. She presented PaO_2_/FIO_2_ 163 mmHg, leukocytosis, and the auscultation of bilateral snoring in the lung bases. On D10 and D11, the patient’s clinical symptoms worsened significantly. She presented ventricular tachycardia, hemodynamic instability (163 BPM), hypotension (83/44 mmHg), PaO_2_/FIO_2_ mmHg, and respiratory distress. The patient was kept in the prone position and kept for 12 h, increasing the PaO_2_/FIO_2_ ratio to 130 mmHg.

On D12, the respiratory conditions and clinical parameters became even worse. The oxygen saturation dropped to 56%, auscultation showed coarse breath sounds, wheezes and coarse crackles. The medical team performed procedures of hemodynamic stabilization, but there was no improvement. The patient progressed to cardiorespiratory arrest and death, despite all attempts of cardiopulmonary resuscitation. Possible causes of progressive clinical worsening leading to cardiac death might be due to continuous stress, overweight condition, and uncertain COVID-19 treatment at the time of hospitalization.

During the mother’s hospitalization, the newborn was fully assisted by the Newborn Intensive Care Unit (NICU) team. The baby’s nasopharyngeal and oropharyngeal secretions were collected, then tested for the same respiratory viral infections tested in her mother. All results were negative, including SARS-CoV-2. During hospitalization in the NICU, the newborn remained asymptomatic without leukocytosis throughout the period. No other treatment was necessary at the NICU. The baby remained under observation in the NICU for eight days until hospital discharge. His follow-up showed no vertical transmission for SARS-CoV-2.

## 3. Results

### 3.1. Specimen Collection and Diagnostic Testing for SARS-CoV-2

The mother’s clinical samples were collected from the nasopharynx and oropharynx, by swabs. The samples from the newborn were collected with mini swabs by the Intensive Care Unit (ICU) professionals and placed in 2 mL of universal viral transport media. Both samples were subjected to RT-PCR assay following the Charité protocol for influenza A and B, parainfluenza, respiratory syncytial virus, rhinovirus, adenovirus, and SARS-CoV-2 as recommended by WHO and by the Brazilian Ministry of Health [8,9]. Testing consisted of extraction of nucleic acid in a QIAamp Diagnostics platform, followed by RT-PCR on Applied Biosystems 7500 Real-Time PCR System [10].

### 3.2. Patient’s Labor and Personal Protective Equipment (PPE)

The emergency cesarean was performed at an operating room close to the ICU and obstetrics/neonatology departments. It was prepared following the safety protocols. The surgical environment and its surfaces were disinfected using 2% chlorhexidine. All healthcare professionals participating in the surgery and postoperatively wore specific individual protective equipment for the prevention of COVID-19, such as face shield, gloves, N95 masks, cape, disposable surgical cap, and shoe covers up to the knees.

The delivery was supported by a team consisting of an obstetrician, pediatrician, and nurses, evolving without obstetric complications due to the distance and protection measures maintained for the newborn. To prevent COVID-19 dissemination, the umbilical cord was clamped immediately, and there was no maternal contact with the newborn. The baby was born at 31 weeks, weighed 1534 g, and Apgar scored 5 in the first and fifth minutes. Afterward, the female baby was placed in an incubator under IVM due to severe respiratory distress resulting from moderate prematurity and sent to the NICU, where she remained under constant observation.

## 4. Discussion

Since the spread of the COVID-19 pandemic, many questions have been raised about SARS-CoV-2 transmission. In this paper, we reported a case of premature labor due to severe COVID-19 illness without vertical transmission [11,12]. The mother, a young woman with heart disease and mitral valve replacement, had a confirmed diagnosis of COVID-19 during her 28th week of pregnancy requiring emergency delivery. The mother did not respond to the treatment because of COVID-19 clinical complications, that evolved to death 12 days after delivering. The baby evolved well with no signs of vertical transmission and negative results for SARS-CoV-2. This case report illustrates some aspects of this emerging viral outbreak that are not yet fully understood, especially about mother-to-child transmission [13,14].

According to the patient and her husband, on her initial evaluation at the hospital, she had no recent national and international travelling history or had not been in contact with people from regions affected by SARS-CoV-2. Although the source of her infection is unknown, her case shows that in Brazil, community transmission was already active by February/March 2020 [15,16]. Initially, pregnant women, as well as cardiopathic, elderly, obese, and diabetics patients were classified as at-risk groups for SARS-CoV-2, which could evolve to unfavorable clinical outcomes. Pregnant women with cardiomyopathy are a special risk group, with little scientific information with regard to the chances of vertical transmission [12,16]. Here, is a case of a severe COVID-19 that evolved to death, suggesting a higher viral load of SARS-CoV-2, and higher risk of vertical transmission [17], no viral transmission was observed.

The literature shows that most pregnant women are asymptomatic or show mild symptoms. Only a minority of cases among pregnant women infected by SARS-CoV-2 required ICU admission and IVM. During the analysis of these studies, there were no reports of vertical transmissions. Currently, there is no clearcut information on how immune regulation is related to pregnancy, how it alters the course of COVID-19, neither how the mother’s immune system inhibits transmission of the virus to the fetus [18].

The risk of vertical transmission is still poorly understood. Up to this date, there has been no firm support for this type of transmission [18,19]. An important result is that the nasopharyngeal swab sample was collected within the first 24 h of the newborn’s life, which showed a negative result for SARS-CoV-2 [18]. It is believed that in addition to the maternal immune system, pre- and trans-operative care in cesarean sections, such as wearing masks by the mother, disinfection of surgical environment, and isolation measures implemented immediately after birth, decreases the chances of contagion of the newborn [19].

Among the relevant information in this case, we called attention to the patient’s initial visit to the emergency department due to a condition of ventricular tachyarrhythmia, when she was hospitalized for the first time for 16 days without signs, or symptoms of any acute or chronic respiratory infection. At that time, testing for SARS-CoV-2 was very scarce in Brazil, and only suspected cases were routinely tested [20]. However, just 5 days after her first hospital discharge, she returned with severe dyspnea, and a high fever of 39 °C, rapidly evolving during her medical care for severe respiratory distress. This case reaffirms the difficulty of predicting the prognosis of patients belonging to the risk groups for COVID-19, who are susceptible to developing the severe and unfavorable clinical complications of the disease. Additionally, is worth emphasizing the importance of monitoring the rules of social distancing, and lockdown during the pandemic, by the government and federal authorities, as cases of community transmission is real [12,13,14,15,16,17,18,19,20].

More than a year since the COVID-19 outbreak, there is still no clear evidence about the best and safest way to deliver babies, and how it can increase or decrease the contagious risk of SARS-CoV-2. Furthermore, SARS-CoV-2 transmission occurs mainly through the air by respiratory droplets, body fluids contact (blood, saliva, and urine), sneezes, coughs, phlegm and interpersonal close contact, followed by contact with mouth, nose, and/or the eyes. During vaginal delivery, it is suggested that the contact of the newborn with maternal body fluids can be a possible risk of transmission. However, some studies have demonstrated SARS-CoV-2 negative results in the amniotic fluid, cord blood, and breast milk. This reinforces the idea of a reduced risk of vertical transmission [21,22,23].

This article describes an unusually severe case of COVID-19 in a pregnant young woman with mitral valve replacement who went into emergency labor during the complication of the disease without vertical transmission. The premature baby evolved well, without immediate or later signs of the SARS-CoV-2 infection. Despite the integrated participation of the team composed of physicians (gynecologist, obstetrician, neonatologist, cardiologist and infectious disease specialist) and nurses (obstetrician, neonatologist and cardiologist) to care both mother and newborn, the mother died of COVID-19 complications despite of the treatment recommendations at the time (March 2020) [21,22,23].

## 5. Conclusions

We assume that the SARS-CoV-2 infection in the young pregnant woman in this study had a fatal development due to her previous clinically extremely vulnerable comorbidities. This includes a combination of pregnancy and cardiomyopathy, which differs from other cases described in the literature that include a healthy women population. It is important to note that, besides the severe and acute COVID-19 course, the baby was delivered by an emergency cesarean and showed no sign of early or late SARS-CoV-2 infection. It corroborates with the lower likelihood of SARS-CoV-2 vertical transmission. More studies are needed to understand how this protection happens.

## Figures and Tables

**Figure 1 viruses-14-00675-f001:**
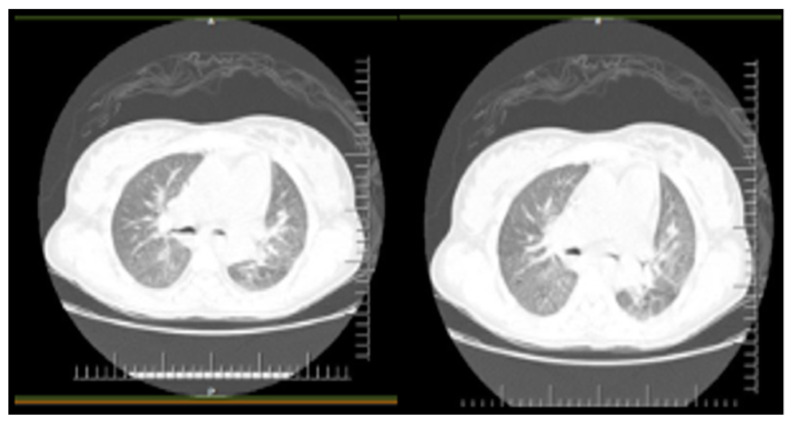
Thoracic computerized tomography showed multiple ground-glass parenchymal opacities, predominantly in the upper lobes of both lungs. Bilateral pleural effusion with parenchymal consolidations. Bilateral multifocal distribution. The extent of pulmonary involvement estimated was greater than 60%. Such findings are consistent with viral pneumonia and differential diagnosis of COVID-19 (31 March 2020, day 2 of the onset of symptoms).

**Table 1 viruses-14-00675-t001:** Daily laboratory results of the pregnant woman with COVID-19 following the cesarean procedure.

Test	References	Days Following Cesarean Procedure									
		D2	D3	D4	D5	D6	D7	D8	D9	D10	D11
		02/04	03/04	04/04	05/04	06/04	07/04	08/04	09/04	10/04	11/04
C reactive protein (mg/dL)	<1.0	0.5	2.0	7.42	4.6	9.0	-	-	15.6	9.0	-
Sodium (mmol/L)	137–145	146	149	150	147	148	-	-	147	146	149
Potassium (mmol/L)	3.5–5.1	3.4	3.2	3.9	2.4	2.7	-	-	3.0	4.1	4.9
pCO_2_ (mmhg)	35–45	76	34	36	32.4	100	-	-	44.4	47	44
pO_2_ (mmhg)	75–100	76	165	191	78.4	107	-	-	56	130	139
Urea (mg/dL)	15–36	25	48	45	26	28	-	-	24	33	-
Creatinine (mg/dL)	0.52–1.04	0.3	0.3	0.2	0.2	0.3	-	-	0.2	0.2	-
Calcium (mg/dL)	8.4–10.2	7.8	7.3	8.0	7.4	-	-	-	7.2	6.9	-
Hemoglobin (g/dL)	12–18	9.1	8.6	8.9	8.6	8.9	-	-	8.5	7.8	-
Hematocrit (%)	36–55	27.7	23.6	27	26.2	28.1	-	-	24.1	25	-
Leukocytes (per mm^3^)	4.000–10.000	9022	6771	11.570	12.030	13.400	13.550	-	10.430	9985	-
Segments (per mm^3^)	5.232	4739	3564	8.625	11.517	-	-	-	-	-	-
Lymphocyte (per mm^3^)	1.160–4.100	3069	1355	1339	1004	1221	-	-	627	1011	-
Lactic acid (mmol/L)	0.5–2.0	2.3	0.6	0.7	0.9	0.7	-	-	1.0	1.5	4.9
Total bilirubin (mg/dL)	0.2–1.3	0.32	0.29	0.25	0.20	0.70	-	-	-	-	-
Platelets (per mm^3^)	150.000–440.000	162.800	161.600	197.900	259.00	-	-	-	233.100	206.300	-
AST^‡^ (U/L)	14–36	71	39	69	-	-	-	-	86	95	-
ALT^†^ (U/L)	<35	23	19	16	-	-	-	-	15	16	-

## Data Availability

Not applicable.

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
