# Peer review of "Severe COVID-19 in Cardiopath Young Pregnant Patient without Vertical Transmission"

_viruses, 2022, doi:10.3390/v14040675_

Round 1

Reviewer 1 Report

In the current study, the authors a case report of severe COVID-19 in cardiopath young pregnant without vertical transmission. Actually, systematic review have suggested vertical transmission of severe acute respiratory syndrome coronavirus 2 is possible and seems to occur in a minority of cases of maternal coronavirus disease 2019 infection. It can provide us with very limited new information about vertical Transmission of SARS-CoV-2, although the  mother’s initial mild symptoms  progressed to severe clinical conditions. Commodities was still thought to play an important role in the occurrence of fatal outcomes.

Author Response

Reply to reviewer #1

1. Concern of the reviewer

• In the current study, the authors a case report of severe COVID-19 in cardiopath young pregnant without vertical transmission. Actually, systematic review have suggested vertical transmission of severe acute respiratory syndrome coronavirus 2 is possible and seems to occur in a minority of cases of maternal coronavirus disease 2019 infection. It can provide us with very limited new information about vertical Transmission of SARS-CoV-2, although the  mother’s initial mild symptoms  progressed to severe clinical conditions. Commodities was still thought to play an important role in the occurrence of fatal outcomes. 

Our response: Dear Reviewer #1, we appreciate your suggestion regarding the manuscript. Yes, nowadays there are evidences proposing a vertical transmission in severe cases, which is discussed in our discussion, however while the case was conducted back in March 2020, there was not evidences of vertical transmission, mainly, because we were trying to understand about SARS-CoV-2 infection. We believe our case report represents a valid asset to scientific literature due to evidence the importance of comorbidity as risk factor to COVID-19 and a non-vertical transmission in COVID-19 severe case. 

Reviewer 2 Report

I appreciate a lot the paper. I find it well wrote and with good idea research. Furthermore, knowledge on SARS CoV2 is ongoing and in my opine it is crucial sharing expierience and good practice and ideas

Below my suggestions

  1. Introduction: updata data on SARS CoV2 wordwilde at the day of resubmission. Furthermore, add how COVID 19 could be an impact, as already shown in other paper on other epidemic, on maternal services. (see and cite Quaglio G, . Impact of Ebola outbreak on reproductive health services in a rural district of Sierra Leone: a prospective observational study. BMJ Open. 2019 Sep 4;9(9):e029093. doi: 10.1136/bmjopen-2019-029093. 
  2. Methods and case presentation: are clear, well wrote and generally well done
  3. Discussion: add the role of team working beetwen gynecologist and infectious diseases specialist is crucial to investigate and well treat this vulnerable patients. The case is very rare, if you can give some suggestion for scientific and clinical community that came from your very interesting paper

Author Response

Reply to reviewer #2

1. Concern of the reviewer             

• Introduction: updata data on SARS CoV2 wordwilde at the day of resubmission. Furthermore, add how COVID 19 could be an impact, as already shown in other paper on other epidemic, on maternal services. (see and cite Quaglio G, . Impact of Ebola outbreak on reproductive health services in a rural district of Sierra Leone: a prospective observational study. BMJ Open. 2019 Sep 4;9(9):e029093. doi: 10.1136/bmjopen-2019-029093. 

Our response: Dear Reviewer #2, we appreciate your suggestion and concern. The text was carefully added. 

Revised text: Page 2, lines 57-60, “This is due to the fact that the COVID-19 pandemic seriously affects the Brazilian health system, especially specialized maternal services, as the availability of structures, medicines and health professionals were focused on efforts to treat COVID-19 leaving on the sidelines maternal health services.” 2. Concern of the reviewer• Methods and case presentation: are clear, well wrote and generally well done. Our response:               Dear Reviewer #2, we appreciate your comment.

  1. Concern of the reviewer
  • Discussion: add the role of team working beetwen gynecologist and infectious diseases specialist is crucial to investigate and well treat this vulnerable patients. The case is very rare, if you can give some suggestion for scientific and clinical community that came from your very interesting paper

Our response: Dear Reviewer #2, we appreciate your suggestion and concern. The text was carefully added. 

Revised text: Page 6, lines 239-242,Despite the integrated participation of the team composed of physicians (gynecologist, obstetrician, neonatologist, cardiologist and infectious disease specialist), nurses (obstetrician, neonatologist and cardiologist) to care both mother and newborn”

Reviewer 3 Report

The paper is well written and has been presented in an extensive way. I congratulate the authors for the quality of the presentation and the detailed  exhibition of clinical findings. I believe it should be added the  CARE guidelines as a reference care-statement.org. The authors meet the criteria of this statement. 

Author Response

Reply to reviewer #3

1. Concern of the reviewer             

• The paper is well written and has been presented in an extensive way. I congratulate the authors for the quality of the presentation and the detailed  exhibition of clinical findings. I believe it should be added the  CARE guidelines as a reference care-statement.org. The authors meet the criteria of this statement. 

Our response: Dear Reviewer #3, we appreciate your comment.